# Infant with Parotid Sialoblastoma and Nevus Sebaceous, Treated with Surgery and Adjuvant Chemotherapy

**DOI:** 10.3390/children10040628

**Published:** 2023-03-28

**Authors:** Radosław Opiła, Sylwia Feszak, Paweł Wawryków, Jarosław Peregud-Pogorzelski

**Affiliations:** Department of Pediatrics, Pediatric Oncology, and Immunology, Pomeranian Medical University, 71-252 Szczecin, Poland

**Keywords:** sialoblastoma, salivary gland tumor, nevus sebaceous, chemotherapy

## Abstract

Sialoblastoma is an extremely rare embryonal tumor derived from salivary gland primordial cells. Treatment usually consists of surgery alone; however, in some cases, chemotherapy is required and is administered with good response. We present a case of a 5-week-old girl diagnosed with a parotid gland tumor and co-existing nevus sebaceous on the face. Initial tumorectomy was microscopically non-radical and histopathology revealed sialoblastoma. The patient received adjuvant chemotherapy consisting of vincristine, actinomycin, and cyclophosphamide. Due to imaging studies being inconclusive regarding response and possible residual disease, a second surgery (total parotidectomy) was performed. The histopathology results showed fields of necrosis in the parotid gland but no neoplastic cells in the material. The patient remains under watchful observation and there is no evidence of relapse 12 months after the second surgery. The adjuvant chemotherapy regimen with vincristine, actinomycin, and cyclophosphamide is a viable option of treatment in children with sialoblastoma.

## 1. Introduction

Sialoblastoma is an embryonal tumor derived from salivary gland primordial cells. Salivary gland malignancies are uncommon in children [1]. In general, most lesions of the salivary glands are non-neoplastic or benign, and hemangioma is the most common finding in children younger than 1 year of age. However, solid tumors in the pediatric population tend to be malignant in over 50% of cases, according to retrospective data [1,2]. Pleomorphic adenomas are the most common benign tumors found in this site, and mucoepidermoid carcinoma accounts for most of the malignant cases [1]. Sialoblastoma is a very rare tumor; to our knowledge, fewer than 100 cases have been described worldwide. The precise epidemiology is difficult to assess, as some previous cases might have been described as other, more common tumors [3,4].

Surgery is the most effective and often solely sufficient treatment for this tumor; however, some cases described to date required chemotherapy. Most of those cases responded well to systemic treatment, indicating sialoblastoma chemo-sensitivity. Local recurrences of the tumor and even metastatic disease were reported, but overall, based on available reports, prognosis tends to be favorable [5,6].

In this report, we present a case of a 5-week-old girl diagnosed with a parotid gland tumor and co-existing nevus sebaceous on the face, who underwent surgical treatment and received adjuvant chemotherapy.

## 2. Case Presentation

### 2.1. Diagnosis

A 5-week-old girl was admitted to our Oncology Department for a palpable tumor in the chin, located near the left mandibular angle. According to her mother, the tumor seemed to grow rapidly. The girl also presented with a large, peculiar nevus on the left cheek of approximately 4 × 4 cm in size (Figure 1), and a smaller nevus of around 1 cm in diameter, located on the chin. She was born as the first child from the first pregnancy of a 28-year-old mother. The mother suffered from chronic hypothyroidism during pregnancy, which was successfully treated. Additional pregnancy and perinatal history was free of complications. Body mass and other newborn measurements were within the normal range. Family history revealed that the mother had papillary thyroid carcinoma that had been treated with thyroidectomy. No other neoplasms or genetic syndromes were reported in the family. After birth, aside from the nevus sebaceous, asymmetry of the face and hairline descending low in the frontal area were observed. The patient was advised to seek genetic consultation; however, she was unable to do so before admission to our Oncology Department.

An initial ultrasound examination revealed an irregular, solid mass 21 × 24 mm in size with small contact with the superficial lobe of the left parotid gland, with a high level of vascularization and small calcifications. Laboratory tests did not indicate any elevation or significant alterations in markers such as beta subunit of human chorionic gonadotropin (beta-HCG), alpha-fetoprotein (AFP), or neuron-specific enolase (NSE).

Magnetic resonance imaging (MRI) of the head and neck revealed a well-defined tumor mass located between the anterior pole of the parotid gland and the posterior part of the musculus masseter, 23 × 21 × 23 mm in size (Figure 2). The lesion seemed to compress but not infiltrate surrounding structures. Restriction of diffusion was observed within the tumor, in addition to contrast enhancement in some parts. There were no other significant findings within the head and neck. There was no apparent connection of the tumor tissue and the visible lesion in the skin of the face.

After consultation with a maxillo-facial surgeon, the patient received indication for tumor resection. Intended macroscopic total resection of the parotid tumor with simultaneous nevus biopsy was performed. The tumor was accessed with a cut encircling the left mandibular angle. The facial nerve was located and dissected using neuromonitoring. The tumor was dissected and removed in one piece.

Histopathology of the tumor revealed the presence of the primitive basal cells organized in lobular and ductal structures with the following immunohistochemical profile: PanCK+, p63+, S100−, EMA+, GFAP−, chromogranin−, synaptophysin−, beta-catenin−, and non-diagnostic WT1. Neoplastic cells were highly polymorphic, with a mitotic index of 23 mitosis/10 high power fields and a focal Ki67-positive fraction of up to 60% of the cells. Small fields of necrosis and angioinvasion were noted in the peripheral parts of the lesion. Neoplastic cells were found at the surgical margin. The diagnosis was sialoblastoma.

The national reference pathologist confirmed this diagnosis.

Histopathological examination of the nevus revealed typical nevus sebaceous histology.

### 2.2. Treatment

After surgery, the patient was reassessed via MRI, which revealed numerous irregular hypointense fields in the superficial lobe of the left parotid gland merging with the musculus masseter and partially encircling the mandibular ramus. In the posterior pole of the same parotid gland, a new lesion 10 × 4 mm in size was observed with restricted diffusion and contrast enhancement. Imaging results suggested non-specific residual disease and likely post-surgical lesions. A computed tomography scan of the chest, abdomen, and pelvis did not reveal any other suspicious focal lesions.

Complete resection of the suspicious lesion without a significant risk of irreversible facial nerve damage or cosmetic mutilation was not possible. Therefore, after discussing therapeutic options with the parents, we decided to initiate systemic treatment, hoping for a reduction in tumor mass. After reviewing the existing literature on sialoblastoma, our team decided to administer chemotherapy using the VAC (vincristine/actinomycin/cyclophosphamide) regimen in accordance with CWS Guidance protocol. Our patient received a total of five courses, with a 4-week interval between day 1 of each cycle, consisting of vincristine, actinomycin D (actinomycin was omitted in the two first courses due to the patient’s age), and cyclophosphamide given on day 1 of each course. Vincristine was also given on days 8 and 15. The treatment was tolerated well; there was no need for blood transfusion, no serious infections, and no apparent signs of neurotoxicity. After two courses, an imaging evaluation showed no significant changes in the lesion.

Another evaluation after four cycles of VAC was comparable with previous imaging, with no signs of progression or regression of the suspicious lesion.

After surgical consultation, concerned about the residual disease, we decided to perform a second, more radical surgery, followed by two additional courses of VAC. Total parotidectomy with facial nerve dissection under neuromonitoring was performed. The same surgical access as before was used. The surgery resulted in paresis of the marginal mandibular and buccal branches of the facial nerve. One course of postoperative chemotherapy was administered. The histopathology results showed fields of necrosis in the parotid gland but no neoplastic cells in the material. We decided to stop further chemotherapy.

### 2.3. Outcome

A control MRI revealed a heterogenous 17 × 22-mm (AP × RL) tissue with contrast enhancement and a focus of restricted diffusion measuring 6 × 9 mm in the place of the resected parotid gland, which was most likely a post-surgical scar, according to the radiologist; however, residual disease could not be excluded. A similar contrast-enhanced focus measuring 4 × 11 mm was observed at the same site in the subcutaneous tissue near the anterior wall of the external acoustic meatus, in addition to a focus 12 × 8 × 15 mm in size in the subcutaneous tissue of the submandibular area. A multi-disciplinary council including surgeons, radiologists, and the local CWS Guidance Study Group center concluded that the probability of the visible MRI changes being residual disease was low, and the patient was placed under cautious observation, without further interventions.

During follow-up, control MRIs of the primary site, in addition to ultrasound imaging of the abdomen and chest X-rays, were performed every 3 months. In the subsequent imaging studies, the previously described post-surgical lesions were stable in size. One of the lesions, which was located in the area of the resected parotid gland, even decreased slightly in size. No signs of evident relapse were noted during 12 months of observation.

Facial nerve function improved slightly over the course of rehabilitation; however, small facial asymmetry is still present.

An initial genetic consultation has taken place, and the patient is awaiting further molecular testing at the time of the publication of this report.

## 3. Discussion

Sialoblastoma is a malignant neoplasm of embryonal origin that is mostly found in young children, especially in the first months of life; however, patients older than 2 years of age, including teenagers, have been described [5]. Dardick et al., after retrospective examination of specimens from adult patients, suggested that some salivary gland tumors in adults present with similar histology [3]. Sialoblastoma is localized to major salivary glands, with most cases originating from the parotid gland, which tends to be the most common site of salivary gland malignancies in general [1]. Total surgical resection is the most effective treatment and often the only treatment necessary [5,6]. In most of the patients described to date who underwent total resection, no recurrences nor metastases were observed, and no further treatment was needed [5]. However, in some cases, usually with unresectable tumors or with subtotal resection, local recurrence, aggressive infiltration of surrounding structures, and even metastases have been observed. In some cases, a second, more radical surgery was the only treatment for local recurrence, and no further steps were taken with good outcomes [5,6]. A minimum of 20 cases described to date were treated with chemotherapy. Most of those cases responded well to the treatment, suggesting that sialoblastoma is a chemosensitive tumor [5,7,8]. Some of the older patients also received radiotherapy, which resulted in a good response [9,10]. Overall, the prognosis of sialoblastoma is favorable. We found three documented sialoblastoma patients who died due to malignancy, one other patient with a rather dismal prognosis (untreated recurrence with metastases) was described as lost to follow-up, and two other neonates died due to sepsis before treatment started [4,5,11,12,13,14]. It is worth noting that many of the reported cases had a short follow-up; therefore, conclusions on long-term outcomes should be made with caution.

Our patient’s initial resection was non-radical, and histopathological examination revealed a very high mitotic index in the lesion. As the imaging assessment indicated that re-operation was not possible without significant mutilation (especially facial nerve palsy), and with our knowledge regarding the chemosensitivity of sialoblastoma, our team decided to administer adjuvant chemotherapy. The initial goal was to reduce the tumor mass, perform less radical re-operation, and prevent metastasis, as the histopathology of our case seemed unfavorable and metastatic cases had been described in the literature. The effectiveness of the chosen VAC regimen had been confirmed in three previously described pediatric cases [5,15].

The imaging reassessments performed after the administration of the second and fourth courses of chemotherapy were difficult to interpret due to technical reasons (one of the MRI studies had to be performed at a different resolution than the rest), as well as the heterogeneous, diffuse, and non-specific characteristics of the lesions. However, there were no apparent signs of tumor regrowth during treatment. The lesion, which we suspected to be the remnant of the tumor, seemed stable in size and morphology. Therefore, we failed to perform less-mutilating re-operation. On histopathological examination of the resected parotid gland, no apparent neoplastic cells were found; however, atypical fields of necrosis suggested possible foci of the neoplasm, which responded to the chemotherapy. Similar, almost purely necrotic foci of sialoblastoma have previously been described in the literature after treatment with the IVAdo regimen [6]. The VAC regimen, according to our team’s experience, is generally well tolerated by infants; it might be considered the chemotherapy of choice in patients with non-resectable or recurring sialoblastoma. Of note, numerous other chemotherapy regimens have been used with success in this malignancy, as Wang et al. listed in their summary; however, most of them were used in a single patient [5].

The uniqueness of our patient’s case results from the co-occurrence of the nevus sebaceous, which was located on the same side of the face and close to the tumor. This type of nevus is characterized as a congenital hamartoma of a pilosebaceous unit and usually presents as a solitary, oval or linear, and well-defined elevated plaque that is yellow or pink in color [16]. Our patient had the linear and disseminated form of this lesion. We found three other cases in the literature in which a similar cutaneous abnormality co-existed with sialoblastoma [17,18,19], and at least one patient who had sebaceous nevus syndrome (a disease with extensive epidermal nevi accompanied by musculoskeletal disorders and neurological deficits) and was diagnosed with parotid adenocarcinoma on the same side of the face as the nevi [20]. The possible link between the two is unclear. Both salivary glands and skin adnexa share common embryological origin, and there are known morphological and genetic links between neoplasms originating from these two types of exocrine glands [21]. Sebaceous nevi are associated with the emergence of secondary neoplasms within them. These tumors are mostly found in adult patients, and most are benign; however, malignant tumors, such as basal cell carcinoma, have been reported [22]. Somatic mutations of HRAS and KRAS, which are well-known proto-oncogenes associated with cell growth and proliferation pathways, have previously been found in epidermal nevi [23,24,25]. It is possible that in patients like in our case, both cutaneous lesions and sialoblastoma can derive from the descendants of the same mutated epidermal precursor cell placed under the influence of a different micro-environment during embryonal development. However, at this time, there are no solid data to prove this hypothesis.

There have been at least eight reported cases of synchronous sialoblastoma and hepatoblastoma. The analysis of one such case conducted by Yang et al. [26] revealed that certain germline mutations might be associated with the development of such a bizarre combination. Since few cases of sialoblastoma have been reported to date, the combination of sialoblastoma cases with co-existing hepatoblastoma is significant, and every patient with sialoblastoma should be carefully assessed for a possible abdominal tumor with proper abdominal imaging and AFP level measurement (sialoblastoma itself can also generate elevated AFP levels) [27].

## 4. Conclusions

Total surgical resection remains the most effective treatment for sialoblastoma. Chemotherapy should be considered in cases with subtotal resection, local recurrence, or metastases. We believe that chemotherapy might help in avoiding mutilating surgery in some patients; however, in those who have already undergone a surgical procedure, proper assessment of the residual disease and its response to treatment might be challenging. A chemotherapy regimen with vincristine, actinomycin, and cyclophosphamide is well tolerated, even by infants, and it appears to be effective against tumor cells in vivo. Some cases of sialoblastoma may co-exist with nevus sebaceus, which may suggest a common pathogenic factor.

## Figures and Tables

**Figure 1 children-10-00628-f001:**
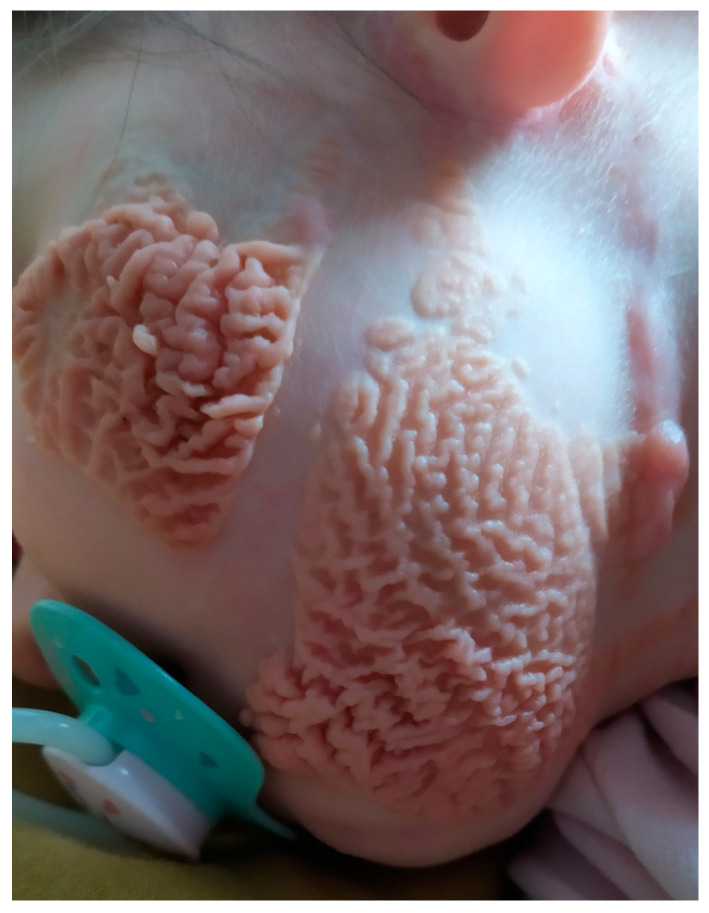
Skin lesion (nevus sebaceous) on the patient’s cheek and tumorectomy scar on the right.

**Figure 2 children-10-00628-f002:**
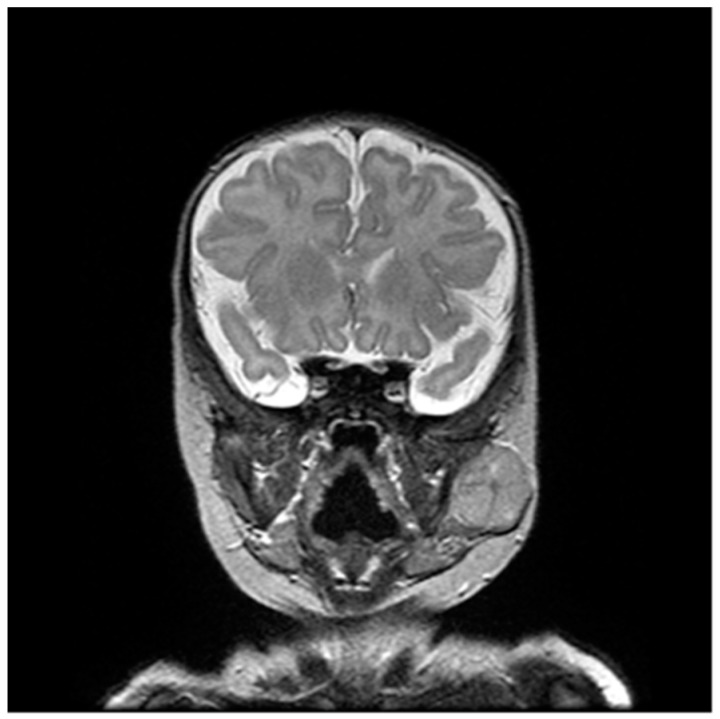
Coronal plane image of the tumor in T2 FSE (Fast Spin Echo) magnetic resonance imaging.

## Data Availability

Not applicable.

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
