# Peer review of "Infant with Parotid Sialoblastoma and Nevus Sebaceous, Treated with Surgery and Adjuvant Chemotherapy"

_children, 2023, doi:10.3390/children10040628_

Round 1
Reviewer 1 Report
Thank you, a very informative and well-written case report of an infrequent entity. There are a few questions to be cleared.
Figure 1. It would be very nice to see a representative MR picture of the lesion. On the right side?
Line 41-41: The authors stated, » palpable tumor in the left submandibular region«. Later in the text, Line 63-65: The MR has shown a mass in the parotid region. Was there such a difference in the clinical examination and the MR, was that something the authors have seen before in such a small child?
Line 68: You stated » no other significant findings within the head«. If the authors meant the intracranial space, I would propose to changing word head to another. If the authors meant the region of the head, I would propose »head and neck«. As it appropriately describes the wider field usually found in MR and the interest zone radiologists and clinicians have.
Line 71: There might be a slight misunderstanding of total macroscopic resection in the first line of surgical treatment. It is also stated in the Introduction as tumorectomy and later in the text as subtotal (Line 155). One option is to go for the intended macroscopic total resection for the first time. That resulted in an R1 resection. As stated in Line 78-79. The last case, it is better to stick to the same because total and subtotal usually surgically mean the anatomical volume of resection of the parotid gland.
Line 91: The authors opted for Chemo since more radical resection would / might damage the facial nerve. Was that the decision of the tumor board or the parents? Sometimes such decisions are tough, especially when multimodal treatment is available or when dealing with infrequent diseases.
Line 103-104: The authors state that no signs of progression were found. But they opted for radical surgery. Could you explain why? I feel this is not a contextual problem but rather information readers would miss.
Author Response
Review 1
- Figure 1: Thank you for the suggestion. We will try to find a representative image and attach it to the manuscript if possible.
- Line 41-41: Interesting remark, that did not catch our attention earlier. The physician who examined the child for the first time in our clinic described the tumor location as submandibular in medical history of the patient. I remember that the mass was indeed palpable near and slightly below mandibular angle. In my opinion, it might might have been the most “prominent” part of the tumor in physical examination. Please remember, that the girl was just 5 weeks old at the time and it was probably difficult to assess the anatomy precisely. If you look at the Figure 1 and newly attached MRI scan picture, the post surgery scar shows more or less the area where the tumor was first visible and palpable.
We believe that we can avoid confusion by changing “submandibular” to “located near mandibular angle”.
- Line 68: Both head (including intracranial space) and neck were assessed in MRI examination. We are going to change that to “head and neck” according to your suggestion.
- Line 71: Thank you for addressing this mistake. We are going to change that for “intended macroscopic total resection” as you proposed and change the “subtotal” for more appropriate term.
- Line 91: Tumor board discussed this decision with parents on separate occasions. We advised them to go for chemotherapy treatment not only because of the possible facial nerve damage, but also because we were concerned about the tumor histology, with its very high mitotic index (as unfolded in “Discussion” section. We informed them, that they can refuse chemotherapy and choose the second surgery right away. After considering both scenarios, they agreed to chemotherapy. We added a short statement in the text to make their involvement in decision making more obvious.
- Line 103-104: We were worried about possible remnants of the disease in the parotid gland and we wanted to avoid another non-radical resection – we unfolded this topic in the “Discussion” section of the original manuscript, as we wanted “Case presentation” section to be concise and easy to read. Parents of the patient were also very concerned about the same issue and opted for total parotidectomy. In revised text, we address that shortly in “Case presentation” section with: “concerned about residual disease” and we add a remark in line 93/94 that states “hoping for reduction of tumor mass”
Reviewer 2 Report
Thank you for entrusting me with a review of the manuscript "Infant with parotid sialoblastoma and nevus sebaceous, treated with surgery and adjuvant chemotherapy". Below are my comments.
Title and abstract
- This section is clear and well structured.
Introduction
- This section should be slightly enriched. I suggest expanding the sentence on treatment to the size of a paragraph and moving the sentence "In our report, we present..." to a separate paragraph.
Case Presentation
- If possible, it is worth replacing Figure 1 with a photo from before the operation. Then, optionally, you can post a photo with a postoperative scar in the further part of the case presentation.
- The entire manuscript should comply with the 2013 CARE Checklist. Therefore, I encourage you to post "Timeline - Historical and current information from this episode of care organized as a timeline (figure or table)" and name the sections and subsections of the manuscript in accordance with the CARE nomenclature.
- It is necessary to describe the course of both operations, including surgical access, the extent of resection, possible intraoperative examinations, difficulties, and complications. Please include photos or drawings if possible. This is of key importance for surgeons who, based on this case report, will be preparing to operate on similar tumors.
- The course of observation concerning the paresis of nerve VII should be described as far as possible, preferably the results of the House-Brackmann assessment should be presented. Were and what steps were taken to reconstruct the nerve?
- Please post the results of the genetic consultation if they appear before the publication of the manuscript.
Discussion
- This section is well structured and, in my opinion, comprehensive.
Conclusions
- I consider the conclusions to be accurate and well-supported.
Back Matter
- The Acknowledgments subsection was not properly populated or deleted.
Author Response
Review 2:
- Introduction:
1) We followed your suggestion and enriched this section with more sentences, keeping it succinct as journal guidelines suggest.
- Case presentation:
1) Unfortunately, we do not have got a good resolution pre-surgery photo that shows the skin lesion in its whole. This is why we have chosen this one. We are adding a representative MRI photo, as other reviewer suggested.
2) Sections and subsections were named and arranged in accordance to journal guidelines for authors. Please note, that even if arranged differently, contents of the whole article overlap quite accurately with CARE checklist.
3) Thank you for addressing the issue. We expanded information about the surgery based on the medical documentation of the patient we own. Please note, however, that authors of this article are not surgeons, and that the first and the second surgery were performed by independent teams in two different centers. We also wanted to focus on describing issues surrounding chemotherapy, as cases like this were reported less than cases with solely surgical treatment.
4) We add more information about the paresis and clinical improvement we observed during follow-up, as you suggested. There were no reconstruction surgical procedures. Patient had undergone rehabilitation. Facial nerve paresis was unfortunately not assessed in this patient using any objective scale at any point of the rehabilitation process.
Round 2
Reviewer 2 Report
The manuscript has been revised according to my comments.